# Structured Reordering for Modeling Latent Alignments in Sequence Transduction

**Bailin Wang**[1]   **Mirella Lapata**[1]   **Ivan Titov**[1,2,3]
[1]University of Edinburgh   [2]University of Amsterdam   [3]Innopolis University
bailin.wang@ed.ac.uk, {mlap, ititov}@inf.ed.ac.uk

## Abstract

Despite success in many domains, neural models struggle in settings where train and test examples are drawn from different distributions. In particular, in contrast to humans, conventional sequence-to-sequence (seq2seq) models fail to generalize systematically, i.e., interpret sentences representing novel combinations of concepts (e.g., text segments) seen in training. Traditional grammar formalisms excel in such settings by implicitly encoding alignments between input and output segments, but are hard to scale and maintain. Instead of engineering a grammar, we directly model segment-to-segment alignments as discrete structured latent variables within a neural seq2seq model. To efficiently explore the large space of alignments, we introduce a reorder-first align-later framework whose central component is a neural reordering module producing *separable* permutations. We present an efficient dynamic programming algorithm performing exact marginal and MAP inference of separable permutations, and, thus, enabling end-to-end differentiable training of our model. The resulting seq2seq model exhibits better systematic generalization than standard models on synthetic problems and NLP tasks (i.e., semantic parsing and machine translation).

## 1   Introduction

Recent advances in deep learning have led to major progress in many domains, with neural models sometimes achieving or even surpassing human performance [49]. However, these methods often struggle in out-of-distribution (*ood*) settings where train and test examples are drawn from different distributions. In particular, unlike humans, conventional sequence-to-sequence (seq2seq) models, widely used in natural language processing (NLP), fail to generalize *systematically* [4, 27, 28], i.e., correctly interpret sentences representing novel combinations of concepts seen in training. Our goal is to provide a mechanism for encouraging systematic generalization in seq2seq models.

To get an intuition about our method, consider the semantic parsing task shown in Figure 1. A learner needs to map a natural language (NL) utterance to a program which can then be executed on a knowledge base. To process the test utterance, the learner needs to first decompose it into two segments previously observed in training (shown in green and blue), and then combine their corresponding program fragments to create a new

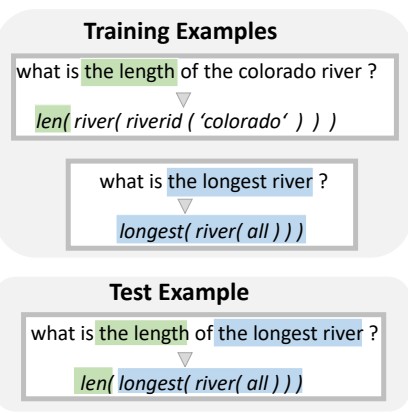

Figure 1: A semantic parser needs to generalize to test examples which contain segments from multiple training examples (shown in green and blue).

in green and blue), and then combine their corresponding program fragments to create a new

35th Conference on Neural Information Processing Systems (NeurIPS 2021).

program. Current seq2seq models fail in this systematic generalization setting [12, 24]. In contrast, traditional grammar formalisms decompose correspondences between utterances and programs into compositional mappings of substructures [46], enabling grammar-based parsers to recombine rules acquired during training, as needed for systematic generalization. Grammars have proven essential in statistical semantic parsing in the pre-neural era [51, 57], and have gained renewed interest now as a means of achieving systematic generalization [18, 41]. However, grammars are hard to create and maintain (e.g., requiring grammar engineering or grammar induction stages) and do not scale well to NLP problems beyond semantic parsing (e.g., machine translation). In this work, we argue that the key property of grammar-based models, giving rise to their improved ood performance, is that a grammar implicitly encodes alignments between input and output segments. For example, in Figure 1, the expected segment-level alignments are 'the length $\rightarrow$ len' and 'the longest river $\rightarrow$ longest(river(all))'. The encoded alignments allow for *explicit decomposition* of input and output into segments, and *consistent mapping* between input and output segments. In contrast, decision rules employed by conventional seq2seq models do not exhibit such properties. For example, recent work [15] shows that primitive units such as words are usually inconsistently mapped across different contexts, preventing these models from generalizing primitive units to new contexts. Instead of developing a full-fledged grammar-based method, we directly model segment-level alignments as structured latent variables. The resulting alignment-driven seq2seq model remains end-to-end differentiable, and, in principle, applicable to any sequence transduction problem.

Modeling segment-level alignments requires simultaneously inducing a segmentation of input and output sequences and discovering correspondences between the input and output segments. While segment-level alignments have been previously incorporated in neural models [50, 55], to maintain tractability, these approaches support only monotonic alignments. The monotonicity assumption is reasonable for certain tasks (e.g., summarization), but it is generally overly restrictive (e.g., consider semantic parsing and machine translation). To relax this assumption, we complement monotonic alignments with an extra reordering step. That is, we first permute the source sequence so that segments within the reordered sequence can be aligned monotonically to segments of the target sequence. Coupling *latent permutations* with monotonic alignments dramatically increases the space of admissible segment alignments.

The space of general permutations is exceedingly large, so, to allow for efficient training, we restrict ourselves to *separable* permutations [5]. We model separable permutations as hierarchical reordering of segments using *permutation trees*. This hierarchical way of modeling permutations reflects the hierarchical nature of language and hence is arguably more appropriate than 'flat' alternatives [33]. Interestingly, recent studies [45, 47] demonstrated that separable permutations are sufficient for capturing the variability of permutations in linguistic constructions across natural languages, providing further motivation for our modeling choice.

Simply marginalizing over all possible separable permutations remains intractable. Instead, inspired by recent work on modeling latent discrete structures [9, 14], we introduce a continuous relaxation of the reordering problem. The key ingredients of the relaxation are two inference strategies: *marginal inference*, which yields the expected permutation under a distribution; *MAP inference*, which returns the most probable permutation. In this work, we propose efficient dynamic programming algorithms to perform *exact* marginal and MAP inference with separable permutations, resulting in effective differentiable neural modules producing relaxed separable permutations. By plugging these modules into an existing module supporting monotonic segment alignments [55], we obtain end-to-end differentiable seq2seq models, supporting non-monotonic segment-level alignments.

In summary, our contributions are:

- A general seq2seq model for NLP tasks that accounts for latent non-monotonic segment-level alignments.

- Novel and efficient algorithms for exact marginal and MAP inference with separable permutations, allowing for end-to-end training using a continuous relaxation.[1]

- Experiments on synthetic problems and NLP tasks (semantic parsing and machine translation) showing that modeling segment alignments is beneficial for systematic generalization.

---

[1]Our code and data are available at `https://github.com/berlino/tensor2struct-public`.

## 2   Background and Related Work

### 2.1   Systematic Generalization

Human learners exhibit systematic generalization, which refers to their ability to generalize from training data to novel situations. This is possible due to the *compositionality* of natural languages - to a large degree, sentences are built using an inventory of primitive concepts and finite structure-building mechanisms [8]. For example, if one understands 'John loves the girl', they should also understand 'The girl loves John' [13]. This is done by 'knowing' the meaning of individual words and the grammatical principle of subject-verb-object composition. As pointed out by Goodwin et al. [15], systematicity entails that primitive units have consistent meaning across different contexts. In contrast, in seq2seq models, the representations of a word are highly influenced by context (see experiments in Lake and Baroni [27]). This is also consistent with the observation that seq2seq models tend to memorize large chunks rather than discover underlying compositional principles [20]. The memorization of large sequences lets the model fit the training distribution but harms out-of-distribution generalization.

### 2.2   Discrete Alignments as Conditional Computation Graphs

Latent discrete structures enable the incorporation of inductive biases into neural models and have been beneficial for a range of problems. For example, input-dependent module layouts [2] or graphs [37] have been explored in visual question answering. There is also a large body of work on inducing task-specific discrete representations (usually trees) for NL sentences [9, 17, 36, 54]. The trees are induced simultaneously with learning a model performing a computation relying on the tree (typically a recursive neural network [43]), while optimizing a task-specific loss. Given the role the structures play in these approaches – i.e., defining the computation flow – we can think of the structures as *conditional computation graphs*.

In this work, we induce discrete alignments as conditional computation graphs to guide seq2seq models. Given a source sequence $x$ with $n$ tokens and a target sequence $y$ with $m$ tokens, we optimize the following objective:

$$\boldsymbol{X} = \text{Encode}_\theta(x) \qquad \mathcal{L}_{\theta,\phi}(x,y) = -\log \mathbb{E}_{p_\phi(\boldsymbol{M}|\boldsymbol{X})} p_\theta(y|\boldsymbol{X}, \boldsymbol{M}) \tag{1}$$

where Encode is a function that embeds $x$ into $\boldsymbol{X} \in \mathbb{R}^{n \times h}$ with $h$ being the hidden size, $\boldsymbol{M} \in \{0,1\}^{n \times m}$ is the alignment matrix between input and output tokens. In this framework, alignments $\boldsymbol{M}$ are separately predicted by $p_\phi(\boldsymbol{M}|\boldsymbol{X})$ to guide the computation $p_\theta(y|\boldsymbol{X}, \boldsymbol{M})$ that maps $x$ to $y$. The parameters of both model components ($\phi$ and $\theta$) are disjoint.

**Relation to Attention**   Standard encoder-decoder models [3] rely on continuous attention weights i.e., $\boldsymbol{M}[:,i] \in \triangle^{n-1}$ for each target token $1 \leq i \leq m$. Discrete versions of attention (aka hard attention) have been studied in previous work [11, 53] and show superior performance in certain tasks. In the discrete case $\boldsymbol{M}$ is a sequence of $m$ categorical random variables. Though discrete, the hard attention only considers word-level alignments, i.e., assumes that each target token is aligned with a single source token. This is a limiting assumption; for example, in traditional statistical machine translation, word-based models (e.g., [6]) are known to achieve dramatically weaker results than phrase-based models (e.g., [26]). In this work, we aim to bring the power of phrase-level (aka segment-level) alignments to neural seq2seq models. [2]

## 3   Latent Segment Alignments via Separable Permutations

Our method integrates a layer of segment-level alignments with a seq2seq model. The architecture of our model is shown in Figure 2. Central to this model is the alignment network, which decomposes the alignment problem into two stages: (i) input reordering and (ii) monotonic alignment between the reordered sequence and the output. Conceptually, we decompose the alignment matrix from Eq 1 into two parts:

$$\boldsymbol{M} = \boldsymbol{M}_{\text{pe}} \boldsymbol{M}_{\text{mo}} \tag{2}$$

---

[2]One of our models (see Section 3.2) still has a flavor of standard continuous attention in that it approximates discrete alignments with continuous expectation.

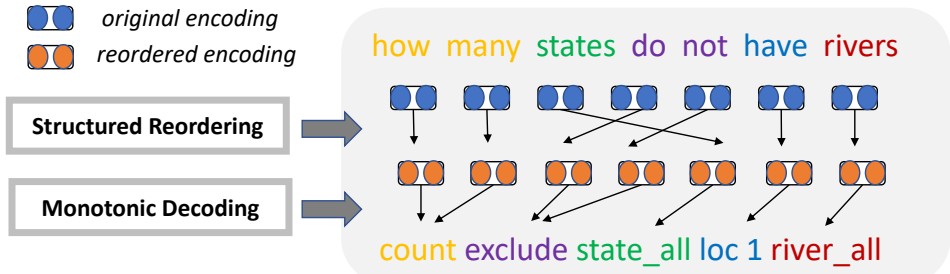

Figure 2: The architecture of our seq2seq model for semantic parsing. After encoding the input utterance, our model permutes the input representations using our reordering module. Then, the reordered encodings will be used for decoding the output program in a monotonic manner.

where $M_{\text{pe}} \in \mathbb{R}^{n \times n}$ is a permutation matrix, and $M_{\text{mo}} \in \mathbb{R}^{n \times m}$ represents monotonic alignments. With this conceptual decomposition, we can rewrite the objective in Eq 1 as follows:

$$\mathcal{L}_{\theta,\phi}(x,y) = -\log \mathbb{E}_{p_\phi(M_{\text{pe}}|x)} \mathbb{E}_{p_{\phi'}(M_{\text{mo}}|M_{\text{pe}}X)} p_\theta(y|M_{\text{pe}}X, M_{\text{mo}}) \tag{3}$$

where $M_{\text{pe}}X$ denotes the reordered representation. With a slight abuse of notation, $\phi$ now denotes the parameters of the model generating permutations, and $\phi'$ denotes the parameters used to produce monotonic alignments. Given the permutation matrix $M_{\text{pe}}$, the second expectation $\mathbb{E}_{p_\phi(M_{\text{mo}}|M_{\text{pe}}X)} p_\theta(y|M_{\text{pe}}X, M_{\text{mo}})$, which we denote as $p_{\theta,\phi'}(y|M_{\text{pe}}X)$, can be handled by existing methods, such as SSNT [55] and SWAN [50]. In the rest of the paper, we choose SSNT as the module for handling monotonic alignment.[3] We can rewrite the objective we optimize in the following compact form:

$$\mathcal{L}_{\theta,\phi,\phi'}(x,y) = -\log \mathbb{E}_{p_\phi(M_{\text{pe}}|x)} p_{\theta,\phi'}(y|M_{\text{pe}}X) \tag{4}$$

## 3.1 Structured Latent Reordering by Binary Permutation Trees

Inspired by Steedman [47], we restrict word reorderings to separable permutations. Formally, separable permutations are defined in terms of binary permutation trees (aka separating trees [5]), i.e., if a permutation can be represented by a permutation tree, it is separable. A binary permutation tree over a permutation of a sequence $1 \ldots n$ is a binary tree in which each node represents the ordering of a segment $i \ldots j$; the children exhaustively split their parent into sub-segments $i \ldots k$ and $k+1 \ldots j$. Each node has a binary label that decides whether the segment of the left child precedes that of the right child.

Bracketing transduction grammar [BTG, 52], which is proposed in the context of machine translation, is the corresponding context-free grammar to represent binary permutation trees. Specifically, BTG has one non-terminal $(X)$ and three anchored rules:

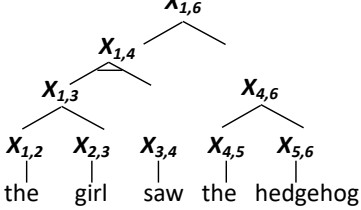

$$\begin{aligned} \mathcal{S}_{i,j,k} &: \quad X_i^k \xrightarrow{Straight} X_i^j X_j^k \\ \mathcal{I}_{i,j,k} &: \quad X_i^k \xrightarrow{Inverted} X_j^k X_i^j \\ \mathcal{T}_i &: \quad X_i^{i+1} \rightarrow x_i \end{aligned}$$

where $X_i^k$ is the anchored non-terminal covering the segment from $i$ to $k$ (excluding $k$). The first two rules decide whether to keep or invert two segments when construct-

Figure 3: The tree represents the reordered sentences 'saw the girl the hedgehog' where $\triangle, \wedge$ denotes *Inverted* and *Straight*, respectively.

ing a larger segment; the last rule states that every word $x_i$ in an utterance is associated with a non-terminal $X_i^{i+1}$. An example is shown in Figure 3. Through this example, we note that the first two rules only signify which segments to inverse; an additional process of interpreting the tree (i.e., performing actual actions of keeping or inverting segments) is needed to obtain the permutated sequence. This hierarchical approach to generating separable permutations reflects the compositional nature of language, and, thus, appears more appealing than using 'flat' alternatives [10, 16, 33]. Moreover, with BTGs, we can incorporate segment-level features to model separable permutations, and design tractable algorithms for learning and inference.

---

[3]In our initial experiments, we found that SWAN works as well as SSNT but is considerably slower.

By assigning a score to each anchored rule using segment-level features, we obtain a distribution over all possible derivations, and use it to compute the objective in Eq 4.

$$p_\phi(D|x) = \frac{\prod_{R \in D} f_\phi(R)}{Z(x, \phi)}, \quad \mathcal{L}_{\theta,\phi,\phi'}(x, y) = -\log \mathbb{E}_{p_\phi(D|x)} p_{\theta,\phi'}(y|M_{\mathrm{pe}}^D X) \quad (5)$$

where $f_\phi$ is a score function assigning a (non-negative) weight to an anchored rule $R \in \{\mathcal{S}, \mathcal{I}, \mathcal{T}\}$, $Z(x, \phi) = \sum_{D'} \prod_{R \in D'} f_\phi(R)$ is the partition function, which can be computed using the inside algorithm, $M_{\mathrm{pe}}^D$ is the permutation matrix corresponding to the derivation $D$. BTG, along with the weight assigned for each rule, is a weighted context-free grammar (WCFG). In this WCFG, the weight is only normalized at the derivation level. As we will see in Algorithm 1, we are interested in normalizing the weight of production rules and converting the WCFG to an equivalent PCFG following Smith and Johnson [42], so that the probability of a derivation can be computed as follows:

$$p_\phi(D|x) = \prod_{R \in D} G_\phi(R) \quad (6)$$

where $G_\phi(R)$ is the weight of the production rule $R$ under the transformed PCFG. The details of the conversion are provided in the Appendix.

The challenge with optimizing the objective in Eq 5 is that the search space of possible derivations is exponential, making the estimation of the gradients with respect to parameters of the reordering component ($\phi$) non-trivial. We now present two differentiable surrogates we use.

## 3.2 Soft Reordering: Computing Marginal Permutations

The first strategy is to use the deterministic expectation of permutations to softly reorder a sentence, analogous to the way standard attention approximates categorical random variables. Specifically, we use the following approximation:

$$M'_{\mathrm{pe}} = \mathbb{E}_{p_\phi(D|x)} M_{\mathrm{pe}}^D$$
$$\mathcal{L}_{\theta,\phi,\phi'}(x, y) \approx -\log p_{\theta,\phi'}(y|M'_{\mathrm{pe}} X)$$

where $M'_{\mathrm{pe}}$ is the marginal permutation matrix, and it can be treated as structured attention [25]. Methods for performing marginal inference for anchored rules, i.e., computing the marginal distribution of production rules are well-known in NLP [32]. However, we are interested in the marginal permutation matrix (or equivalently the expectation of the matrix components) as the matrix is the data structure that is ultimately used in our model. As a key contribution of this work, we propose an efficient algorithm to exactly compute the marginal permutation matrix using dynamic programming.

In order to compute the marginal permutation matrix we need to marginalize over the exponentially many derivations of each permutation.

---

**Algorithm 1** Dynamic programming for computing marginals and differentiable sampling of permutation matrix wrt. a parameterized grammar

**Input:** $G_\phi(R)$: probability of an anchored rule $R$
$\quad\quad\quad$ *sampling*: whether perform sampling

1: **for** $i := 1$ **to** $n$ **do**
2: $\quad\quad E_i^{i+1} = 1$
3: **end for**
4: **for** $w := 2$ **to** $n$ **do** $\quad\quad\quad$ ▷ width of spans
5: $\quad\quad$ **for** $i := 1$ **to** $n - w + 1$ **do**
6: $\quad\quad\quad\quad k := i + w$
7: $\quad\quad\quad\quad$ **if** *sampling* **then**
8: $\quad\quad\quad\quad\quad\quad \hat{G}_\phi(R) = \mathrm{s\_arg\,max}(G_\phi(R))$
9: $\quad\quad\quad\quad$ **else** $\quad\quad\quad$ ▷ computing marginals
10: $\quad\quad\quad\quad\quad\quad \hat{G}_\phi(R) = G_\phi(R)$
11: $\quad\quad\quad\quad$ **end if**
12: $\quad\quad\quad\quad$ **for** $j := i + 1$ **to** $k - 1$ **do**
13: $\quad\quad\quad\quad\quad\quad E_i^k \mathrel{+}= \hat{G}_\phi(\mathcal{S}_{i,j,k})(E_i^j \oplus E_j^k)$
14: $\quad\quad\quad\quad\quad\quad E_i^k \mathrel{+}= \hat{G}_\phi(\mathcal{I}_{i,j,k})(E_i^j \ominus E_j^k)$
15: $\quad\quad\quad\quad$ **end for**
16: $\quad\quad$ **end for**
17: **end for**
18: **return** $E_1^{n+1}$

---

We propose to map a derivation of BTG into its corresponding permutation matrix in a recursive manner. Specifically, we first associate word $i$ with an identity permutation matrix $M_i^{i+1} = 1$; then we associate *Straight* and *Inverted* rules with direct $\oplus$ and skew $\ominus$ sums of permutation matrices, respectively:

$$A \oplus B = \begin{bmatrix} A & 0 \\ 0 & B \end{bmatrix} \quad\quad A \ominus B = \begin{bmatrix} 0 & A \\ B & 0 \end{bmatrix}$$

For example, the permutation matrix of the derivation tree shown in Figure 3 can be obtained by:

$$M_1^6 = \left( ((M_1^2 \oplus M_2^3) \ominus M_3^4) \oplus (M_4^5 \oplus M_5^6) \right) \quad (7)$$

Intuitively, the permutation matrix of long segments can be constructed by composing permutation matrices of short segments. Motivated by this, we propose a dynamic programming algorithm, which takes advantage of the observation that we can reuse the permutation matrices of short segments when computing permutation matrices of long segments, as shown in Algorithm 1. While the above equation is defined over discrete permutation matrices encoding a single derivation, the algorithm applies recursive rules to expected permutation matrices. Central to the algorithm is the following recursion:

$$\boldsymbol{E}_i^k = \sum_{i<j<k} G_\phi(\mathcal{S}_{i,j,k})(\boldsymbol{E}_i^j \oplus \boldsymbol{E}_j^k) + G_\phi(\mathcal{I}_{i,j,k})(\boldsymbol{E}_i^j \ominus \boldsymbol{E}_j^k) \tag{8}$$

where $\boldsymbol{E}_i^k$ is the expected permutation matrix for the segment from $i$ to $k$, $G_\phi(R)$ is the probability of employing the production rule $R$, defined in Eq 6. Overall, Algorithm 1 is a bottom-up method that constructs expected permutation matrices incrementally in Step 13 and 14, while relying on the probability of the associated production rule. We prove the correctness of this algorithm by induction in the Appendix.

### 3.3 Hard Reordering: Gumbel-Permutation by Differentiable Sampling

During **inference**, for efficiency, it is convenient to rely on the most probable derivation $D'$ and its corresponding most probable $y$:

$$\arg\max_y p_{\theta,\phi'}(y|\boldsymbol{M}_{\text{pe}}^{D'}\boldsymbol{X}) \tag{9}$$

where $D' = \arg\max_D p_\phi(D|x)$. The use of discrete permutations $\boldsymbol{M}_{\text{pe}}^{D'}$ during inference and soft reorderings during training lead to a training-inference gap which may be problematic. Inspired by recent Gumbel-Softmax operator [21, 31] that relaxes the sampling procedure of a categorical distribution using the Gumbel-Max trick, we propose a differentiable procedure to obtain an approximate sample $\boldsymbol{M}_{\text{pe}}^D$ from $p(D|x)$. Concretely, the Gumbel-Softmax operator relaxes the perturb-and-MAP procedure [39], where we add noises to probability logits and then relax the MAP inference (i.e., $\arg\max$ in the categorical case); we denote this operator as $\text{s\_arg}\max$. In our structured case, we perturb the logits of the probabilities of production rules $G_\phi(R)$, and relax the structured MAP inference for our problem. Recall that $p(D|x)$ is converted to a PCFG, and MAP inference for PCFG is algorithmically similar to marginal inference. Intuitively, for each segment, instead of marginalizing over all possible production rules in marginal inference, we choose the one with the highest probability (i.e., a local MAP inference with categorical random variables) during MAP inference. By relaxing each local MAP inference with Gumbel-Softmax (Step 8 of Algorithm 1), we obtain a differentiable sampling procedure. [4] We choose Straight-Through Gumbel-Softmax so that the return of Algorithm 1 is a discrete permutation matrix, and in this way we close the training-inference gap faced by soft reordering.

**Summary** We propose two efficient algorithms for computing marginals and obtaining samples of separable permutations with their distribution parameterized via BTG. In both algorithms, PCFG plays an important role of decomposing a global problem into sub-problems, which explains why we convert $p(D|x)$ into a PCFG in Eq 6. Relying on the proposed algorithms, we present two relaxations of the discrete permutations that let us induce latent reorderings with end-to-end training. We refer to the resulting system as *ReMoto*, short for a seq2seq model with Reordered-then-Monotone alignments. Soft-*ReMoto* and Hard-*ReMoto* denote the versions which use soft marginal permutations and hard Gumbel permutations, respectively.

**Segment-Level Alignments** Segments are considered as the basic elements being manipulated in our reordering module. Concretely, permutation matrices are constructed by hierarchically reordering input segments. SSNT, which is the module on top of our reordering module for monotonically generating output, conceptually also considers segments as basic elements. Intuitively, SSNT alternates between consuming an input segment and generating an output segment. Modeling segments provides a strong inductive bias, reflecting the intuition that sequence transduction in NLP can be largely accomplished by manipulations at the level of segments. In contrast, there is no explicit notion of segments in conventional seq2seq methods.

---

[4]If we change $\text{s\_arg}\max$ with $\arg\max$ in Step 8 of Algorithm 1, we will obtain the algorithm for exact MAP inference.

| Dataset | Input | Output |
|---------|-------|--------|
| Arithmetic | $((1+9)*((7+8)/4))$ | $((19+)((78+)4/)*)$ |
| SCAN-SP | jump twice after walk around left thrice | after (twice (jump), thrice(walk (around, left))) |
| GeoQuery | how many states do not have rivers ? | count(exclude(state(all), loc_1(river(all)))) |

Table 1: Examples of input-output pairs for parsing tasks.

| | Arithmetic | | SCAN-SP | |
|-------|------|------|------|------|
| Model | IID | LEN | IID | LEN |
| Seq2Seq | 100.0 | 0.0 | 100.0 | 13.9 |
| LSTM-based Tagging | 100.0 | 20.6 | 100.0 | 57.7 |
| Sinkhorn-Attention Tagging | 99.5 | 8.8 | 100.0 | 48.2 |
| Soft-*ReMoto* | 100.0 | **86.9** | 100.0 | 100.0 |
|   - *shared parameters* | 100.0 | 40.9 | 100.0 | 100.0 |
| Hard-*ReMoto* | 100.0 | 83.3 | 100.0 | 100.0 |

Table 2: Accuracy (%) on the arithmetic and SCAN-SP tasks.

However, different from our reordering module where segments are first-class objects during modeling, the alternating process of SSNT is realized by a series of token-level decisions (e.g., whether to keep consuming the next input token). Thus, properties of segments (e.g., segment-level features) are not fully exploited in SSNT. In this sense, one potential way to further improve *ReMoto* is to explore better alternatives to SSNT that can treat segments as first-class objects as well. We leave this direction for future work.

**Reordering in Previous Work**    In traditional statistical machine translation (SMT), reorderings are typically handled by a distortion model [e.g., 1] in a pipeline manner. Neubig et al. [35], Nakagawa [34] and Stanojević and Sima'an [44] also use BTGs for modeling reorderings. Stanojević and Sima'an [44] go beyond binarized grammars, showing how to support 5-ary branching permutation trees. Still, they assume the word alignments have been produced on a preprocessing step, using an alignment tool [38]. Relying on these alignments, they induce reorderings. Inversely, we rely on latent reordering to induce the underlying word and segment alignments.

Reordering modules have been previously used in neural models, and can be assigned to the following two categories. First, reordering components [7, 19] were proposed for neural machine translation. However, they are not structured or sufficiently constrained in the sense that they may produce invalid reorderings (e.g., a word is likely to be moved to more than one new position). In contrast, our module is a principled way of dealing with latent reorderings. Second, the generic permutations (i.e., one-to-one matchings or sorting), though having differentiable counterparts [10, 16, 33], do not suit our needs as they are defined in terms of tokens, rather than segments. For comparison, in our experiments, we design baselines that are based on Gumbel-Sinkhorn Network [33], which is used previously in NLP (e.g., [30]).

## 4   Experiments

First, we consider two diagnostic tasks where we can test the neural reordering module on its own. Then we further assess our general seq2seq model *ReMoto* on two real-world NLP tasks, namely semantic parsing and machine translation.

### 4.1   Diagnostic Tasks

**Arithmetic**    We design a task of converting an arithmetic expression in infix format to the one in postfix format. An example is shown in Table 1. We create a synthetic dataset by sampling data from a PCFG. In order to generalize, a system needs to learn how to manipulate internal sub-structures (i.e., segments) while respecting well-formedness constraints. This task can be solved by the shunting-yard algorithm but we are interested to see if neural networks can solve it and generalize ood by learning from raw infix-postfix pairs. For standard splits (IID), we randomly sample 20k infix-postfix pairs whose nesting depth is set to be between 1 and 6; 10k, 5k, 5k of these pairs are used as train, dev and test sets, respectively. To test systematic generalization, we create a Length split (LEN) where

training and dev examples remain the same as IID splits, but test examples have a nesting depth of 7. In this way, we test whether a system can generalize to unseen longer input.

**SCAN-SP** We use the SCAN dataset [27], which consists of simple English commands coupled with sequences of discrete actions. Here we use the semantic parsing version, SCAN-SP [18], where the goal is to predict programs corresponding to the action sequences. An example is shown in Table 1. As in these experiments our goal is to test the reordering component alone, we remove parentheses and commas in programs. For example, the program `after (twice (jump), thrice(walk (around, left)))` is converted to a sequence: `after twice jump thrice walk around left`. In this way, the resulting parentheses-free sequence can be viewed as a reordered sequence of the NL utterance 'jump twice after walk around left thrice'. The grammar of the programs is known so we can reconstruct the original program from the intermediate parentheses-free sequences using the grammar. Apart from the standard split (IID, aka simple split [27]), we create a Length split (LEN) where the training set contains NL utterances with a maximum length 5, while utterances in the dev and test sets have a minimum length of 6.[5]

**Baselines and Results** In both diagnostic tasks, we use *ReMoto* with a trivial monotonic alignment matrix $M_{mo}$ (an identity matrix) in Eq 3. Essentially, *ReMoto* becomes a sequence tagging model. We consider three baselines: (1) vanilla Seq2Seq models with Luong attention [29]; (2) an LSTM-based tagging model which learns the reordering implicitly, and can be viewed as a version *ReMoto* with a trivial $M_{pe}$ and $M_{mo}$; (3) Sinkhorn Attention that replaces the permutation matrix of Soft-*ReMoto* in Eq 4 by Gumbel-Sinkhorn networks [33].

We report results by averaging over three runs in Table 2. In both datasets, almost all methods achieve perfect accuracy in IID splits. However, baseline systems cannot generalize well to the challenging LEN splits. In contrast, our methods, both Soft-*ReMoto* and Hard-*ReMoto*, perform very well on LEN splits, surpassing the best baseline system by large margins ($> 40\%$). The results indicate that *ReMoto*, particularly its neural reordering module, has the right inductive bias to learn reorderings. We also test a variant Soft-*ReMoto* where parameters $\theta, \phi$ with shared input embeddings. This variant does not generalize well to the LEN split on the arithmetic task, showing that it is beneficial to split models of the 'syntax' (i.e., alignment) and 'semantics', confirming what has been previously observed [17, 40].

## 4.2 Semantic Parsing

Our second experiment is on semantic parsing where *ReMoto* models the latent alignment between NL utterances and their corresponding programs. We use GeoQuery dataset [56] which contains 880 utterance-programs pairs. The programs are in variable-free form [23]; an example is shown in Table 1.[6] Similarly to SCAN-SP, we transform the programs into parentheses-free form which have better structural correspondence with utterances. Again, we can reconstruct the original programs based on the grammar. An example of such parentheses-free form is shown in Figure 2. Apart from the standard version, we also experiment with the Chinese and German versions of GeoQuery [22, 48]. Since different languages exhibit divergent word orders [47], the results in the multilingual setting will tell us if our model can deal with this variability.

In addition to standard IID splits, we create a LEN split where the training examples have parentheses-free programs with a maximum length 4; the dev and test examples have programs with a minimum length 5. We also experiment with the TEMP split [18] where training and test examples have programs with disjoint templates.

**Baselines and Results** Apart from conventional seq2seq models, for comparison, we also implemented the syntactic attention [40]. Our model *ReMoto* is similar in spirit to the syntactic attention, 'syntax' in their model (i.e., alignment) and 'semantics' (i.e., producing the representation relying

---

[5]Since we use the program form, the original length split [27], which is based on the length of action sequence, is not very suitable in our experiments.

[6]We use the varaible-free form, as opposed to other alternatives such lambda calculus, for two reasons: 1) variable-free programs have been commonly used in systematic generalization settings [18, 41], probably it is easier to construct generalization splits using this form; 2) the variable-free form is more suitable for modeling alignments since variables in programs usually make alignments hard to define.

| Model | EN | | | ZH | | | DE | | |
|---|---|---|---|---|---|---|---|---|---|
| | IID | TEMP | LEN | IID | TEMP | LEN | IID | TEMP | LEN |
| Seq2Seq | 75.7 | 38.8 | 21.8 | 72.5 | 25.4 | 19.8 | 56.1 | 18.8 | 15.2 |
| Syntactic Attention [40] | 74.3 | 39.1 | 18.3 | 70.2 | 27.9 | 18.7 | 54.3 | 19.3 | 14.2 |
| SSNT [55] | 75.3 | 38.7 | 19.1 | 71.6 | 23.8 | 17.8 | 55.2 | 19.8 | 14.1 |
| Soft-*ReMoto* | 74.5 | 39.3 | 19.8 | 73.4 | 30.3 | 17.3 | 55.8 | 19.5 | 13.4 |
| Hard-*ReMoto* | 75.2 | 43.2 | 23.2 | 74.3 | 45.7 | 22.3 | 55.6 | 22.3 | 16.6 |

Table 3: Exact-match accuracy (%) on three splits of the multilingual GeoQuery dataset. Numbers underlined are significantly better than others (p-value $\leq 0.05$ using the paired permutation test).

on the alignment) are separately modeled. In contrast to our structured mechanism for modeling alignments, their syntactic attention still relies on the conventional attention mechanism. We also compare with SSNT, which can be viewed as an ablated version of *ReMoto* by removing our reordering module.

Results are shown in Table 3. For the challenging TEMP and LEN splits, our best performing model Hard-*ReMoto* achieves consistently stronger performance than seq2seq, syntactic attention and SSNT. Thus, our model bridges the gap between conventional seq2seq models and specialized state-of-the-art grammar-based models [18, 41].[7]

## 4.3 Machine Translation

Our final experiment is on small-scale machine translation tasks, where *ReMoto* models the latent alignments between parallel sentences from two different languages. To probe systematic generalization, we also create a LEN split for each language pair in addition to the standard IID splits.

**English-Japanese**   We use the small en-ja dataset extracted from TANKA Corpus. The original split (IID) has 50k/500/500 examples for train/dev/test with lengths 4-16 words.[8] We create a LEN split where the English sentences of training examples have a maximum length 12 whereas the English sentences in dev/test have a minimum length 13. The LEN split has 50k/538/538 examples for train/dev/test, respectively.

**Chinese-English**   We extract a subset from FBIS corpus (LDC2003E14) by filtering English sentences with length 4-30. We randomly shuffle the resulting data to obtain an IID split which has 141k/3k/3k examples for train/dev/test, respectively. In addition, we create a LEN split where English sentences of training examples have a maximum length 29 whereas the English sentences of dev/test examples have a length 30. The LEN split has 140k/4k/4k examples as train/dev/test sets respectively.

**Baselines and Results**   In addition to the conventional seq2seq, we compare with the original SSNT model which only accounts for monotonic alignments. We also implemented a variant that combines SSNT with the local reordering module [19] as our baseline to show the advantage of our structured ordering module.

| | EN-JA | | ZH-EN | |
|---|---|---|---|---|
| | IID | LEN | IID | LEN |
| Seq2Seq | 35.6 | 25.3 | 21.4 | 18.1 |
| SSNT [55] | 36.3 | 26.5 | 20.5 | 17.3 |
| Local Reordering [19] | 36.0 | 27.1 | 21.8 | 17.8 |
| Soft-*ReMoto* | 36.6 | 27.5 | 22.3 | 19.2 |
| Hard-*ReMoto* | **37.4** | **28.7** | **22.6** | **19.5** |

Table 4: BLEU scores on the EN-JA and ZH-EN translation.

Results are shown in Table 4. Our model, especially Hard-*ReMoto*, consistently outperforms other baselines on both splits. In EN-JA translation, the advantage of our best-performance Hard-*ReMoto* is slightly more pronounced in the LEN split than in the IID split. In ZH-EN translation, while SSNT and its variant do not outperform seq2seq in the LEN split, *ReMoto* can still achieve better results than seq2seq. These results show that our model is better than its alternatives at generalizing to longer sentences for machine translation.

---

[7]NQG [41] achieves 35.0% in the English LEN, and SBSP [18] (without lexicon) achieves 65.9% in the English TEMP in execution accuracy. Both models are augmented with pre-trained representations (BERT).

[8]https://github.com/odashi/small_parallel_enja

| | |
|---|---|
| original input: | 在$^1_{\text{in}}$ 美国$^2_{\text{usa}}$ 哪些$^3_{\text{which}}$ 州$^4_{\text{state}}$ 与$^5$ 最长$^6_{\text{longest}}$ 的$^7$ 河流$^8_{\text{river}}$ 接壤$^9_{\text{border}}$ |
| reordered input: | 州$^4_{\text{state}}$ 接壤$^9_{\text{border}}$ 最长$^6_{\text{longest}}$ 的$^7$ 河流$^8_{\text{river}}$ 与$^5$ 哪些$^3_{\text{which}}$ 美国$^2_{\text{usa}}$ 在$^1_{\text{in}}$ |
| prediction: | `state`$^4$ `next_to_2`$^9$ `longest river`$^{6,7,8}$ `loc_2 countryid_ENTITY`$^{5,3,2}$ |
| ground truth: | `state next_to_2 longest river loc_2 countryid_ENTITY` |
| original input: | according$^1$ to$^2$ the$^3$ newspaper$^4$ ,$^5$ there$^6$ was$^7$ a$^8$ big$^9$ fire$^{10}$ last$^{11}$ night$^{12}$ |
| reordered input: | according$^1$ to$^2$ the$^3$ newspaper$^4$ ,$^5$ night$^{12}$ last$^{11}$ big$^9$ fire$^{10}$ a$^8$ there$^6$ was$^7$ |
| prediction: | 新によれば、$^{1,2,3,4,5}$ 昨夜$^{12}$ 大$^{11,9}$ 火事$^{10}$ があ$^{8,6}$ った$^7$ |
| ground truth: | 新によると昨夜大火事があった |

Table 5: Output examples of Chinese semantic parsing and English-Japanese translation. For clarity, the input words are labeled with position indices, and, for semantic parsing, with English translations. A prediction consists of multiple segments, each annotated with a superscript referring to input tokens.

**Interpretability** Latent alignments, apart from promoting systematic generalization, also lead to better interpretability as discrete alignments reveal the internal process for generating output. For example, in Table 5, we show a few examples from our model. Each output segment is associated with an underlying rationale, i.e. a segment of the reordered input.

## 5 Conclusion and Future Work

In this work, we propose a new general seq2seq model that accounts for latent segment-level alignments. Central to this model is a novel structured reordering module which is coupled with existing modules to handle non-monotonic segment alignments. We model reorderings as separable permutations and propose an efficient dynamic programming algorithm to perform marginal inference and sampling. It allows latent reorderings to be induced with end-to-end training. Empirical results on both synthetic and real-world datasets show that our model can achieve better systematic generalization than conventional seq2seq models.

The strong inductive bias introduced by modeling alignments in this work could be potentially beneficial in weakly-supervised and low-resource settings, such as weakly-supervised semantic parsing and low-resource machine translation where conventional seq2seq models usually do not perform well.

## Acknowledgements

We thank Miloš Stanojević and Khalil Sima'an for their valuable comments; Lei Yu and Chris Dyer for providing the preprocessed data for machine translation; the anonymous reviewers for their helpful feedback. We gratefully acknowledge the support of the European Research Council (Titov: ERC StG BroadSem 678254; Lapata: ERC CoG TransModal 681760) and the Dutch National Science Foundation (NWO VIDI 639.022.518). Titov is also supported by the Analytical Center for the Government of Russian Federation (agreement 70-2021-00143, dd. 01.11.2021, IGK 000000D730321P5Q0002).

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
