# Appendix: Structured Reordering for Modeling Latent Alignments in Sequence Transduction

**WCFG to PCFG Conversion**   The algorithm of converting a WCFG to its equivalent PCFG is shown in Algorithm 1. In a bottom-up manner, the algorithm first computes an inner weight $\beta[X_i^k]$ for each segment, which is the total weight of all derivations with root $X_i^k$. Then the algorithm normalizes the weight of production rules whose left-hand side is $X_i^k$ using the inner weight. The resulting normalized weight for a production rule, e.g., $G[X_i^k \xrightarrow{S} X_i^j X_j^k]$, is the conditional probability of applying the rule $X_i^k \xrightarrow{S} X_i^j X_j^k$ given the presence of the segment $X_i^k$. The PCFG is equivalent to the original WCFG in the sense that for each derivation $D$, we have

$$p_\phi(D|x) = \frac{\prod_{R \in D} f_\phi(R)}{Z(x, \phi)} = \prod_{R \in D} G_\phi(R)$$

where $Z(x, \phi) = \sum_{D'} \prod_{R \in D'} f_\phi(R)$. Full proof of this equivalence can be found in Smith and Johnson [1]. The factorization of the derivation-level probability to rule-level probability facilitates our design of dynamic programming for marginal inference.

**Proof of the Dynamic Programming for Marginal Inference**   We prove the correctness of the dynamic programming algorithm for computing the marginal permutation matrix of separable permutations by induction as follows.

*Proof.* As a base case, each word (i.e., segment with length 1) is associated with an identity permutation matrix **1**. Then we assume that the marginal permutation matrix for all segments with length $1 < k - i < n$ is $\boldsymbol{E}_i^k$, which is defined as $\mathbb{E}_{p(D_i^k)}[\boldsymbol{M}(D_i^k)]$ where $D_i^k$ is the derivation tree of segment $i$ to $k$, and $\boldsymbol{M}(D_i^k)$ is the permutation matrix corresponding to $D_i^k$. It is obvious that

**Algorithm 1** Converting WCFG to PCFG

---

1: initialize $\beta[\dots]$ to 0
2: **for** $i := 0$ **to** $n - 1$ **do**            ▷ width-1 spans
3:     $\beta[X_i^{i+1}] = 1$
4: **end for**
5: **for** $w := 2$ **to** $n$ **do**            ▷ width of spans
6:     **for** $i := 0$ **to** $n - w$ **do**            ▷ start point
7:         $k := i + w$            ▷ end point
8:         **for** $j := i + 1$ **to** $k - 1$ **do**            ▷ compute inner weight
9:             $\beta[X_i^k]+ = f_\phi(X_i^k \xrightarrow{S} X_j^j X_j^k)\beta[X_i^j]\beta[X_j^k]$            ▷ S: Straight
10:            $\beta[X_i^k]+ = f_\phi(X_i^k \xrightarrow{I} X_i^j X_j^k)\beta[X_i^j]\beta[X_j^k]$            ▷ I: Inverted
11:         **end for**
12:         **for** $j := i + 1$ **to** $k - 1$ **do**            ▷ normalize weight
13:            $G(X_i^k \xrightarrow{S} X_i^j X_j^k) = \frac{f_\phi(X_i^k \xrightarrow{S} X_i^j X_j^k)\beta[X_i^j]\beta[X_j^k]}{\beta[X_i^k]}$
14:            $G(X_i^k \xrightarrow{I} X_i^j X_j^k) = \frac{f_\phi(X_i^k \xrightarrow{I} X_i^j X_j^k)\beta[X_i^j]\beta[X_j^k]}{\beta[X_i^k]}$
15:         **end for**
16:     **end for**
17: **end for**
18: **return** $G[\dots]$

---

$\boldsymbol{E}_i^{i+1} = \boldsymbol{1}$. The marginal permutation matrix for all segments with length $n$ can be obtained by

$$
\begin{aligned}
\boldsymbol{E}_i^k &= \mathbb{E}_{p(D_i^k)}[\boldsymbol{M}(D_i^k)] \\
&= \sum_{i<j<k} \Big( G_\phi(\mathcal{S}_{i,j,k})\big(\mathbb{E}_{p(D_i^j)}[\boldsymbol{M}(D_i^j)]\big) \oplus \mathbb{E}_{p(D_j^k)}[\boldsymbol{M}(D_j^k)]\big) \\
&\quad\quad + G_\phi(\mathcal{I}_{i,j,k})\big(\mathbb{E}_{p(D_i^j)}[\boldsymbol{M}(D_i^j)] \ominus \mathbb{E}_{p(D_j^k)}[\boldsymbol{M}(D_j^k)]\big)\Big) \\
&= \sum_{i<j<k} \Big( G_\phi(\mathcal{S}_{i,j,k})(\boldsymbol{E}_i^j \oplus \boldsymbol{E}_j^k) + G_\phi(\mathcal{I}_{i,j,k})(\boldsymbol{E}_i^j \ominus \boldsymbol{E}_j^k)\Big)
\end{aligned}
$$

where in the second step we consider all the possible expansions of the derivation tree $D_i^k$; in the third step, we obtain the recursion that is used in Step 12-14 of Algorithm 1 by reusing the marginal permutations matrices of shorter segments.   □

**Architecture and Hyperparameters**    The detailed architecture of *ReMoto* is shown in Figure 1. In the structured reordering module, we compute the scores for BTG production rules using span

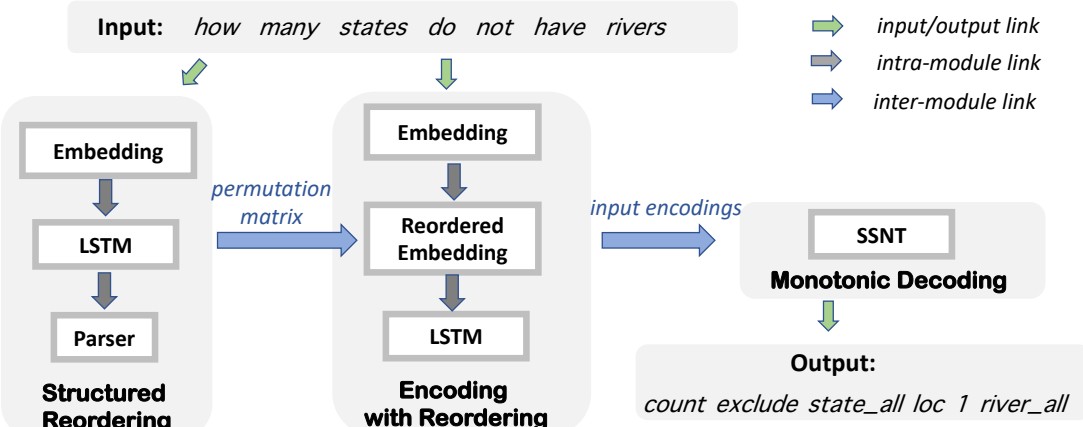

Figure 1: The detailed architecture of our seq2seq model for semantic parsing (view in color). First, the structured reordering module genearates a (relaxed) permutation matrix given the input utterrance. Then, the encoding module generates the representations of the input utterance based on the reordered embeddings, which are computed based on the original embedding and the permutation matrix computed in the first step. Finally, the decoding module, namely SSNT, generates the output program monotonically based on the input encodings.

embeddings [2] followed by a multi-layer perceptron. Specifically, the score function for each rule has form $G(R_{i,j,k}) = \mathrm{MLP}(\boldsymbol{s}_{ij}, \boldsymbol{s}_{jk})$, where $\boldsymbol{s}_{ij}$ and $\boldsymbol{s}_{jk}$ are the span embeddings based on [2], MLP is a multi-layer perceptron that outputs a 2-d vector, which corresponds to the score of $R$=Straight and $R$=Inverted, respectively. Similar to a conventional LSTM-based encoder-decoder model, LSTMs used in structured reordering and encoding module are bidirectional whereas the LSTM for decoding (within SSNT) is unidirectional. We implemented all models using Pytorch [3]. We list the main hyperparameters we tuned are shown in Table 1. The full hyperparameters for each experiment will be released along with the code.

**Training Strategy** Empirically, we found that during training the structured reordering module tends to converge to a sub-optimal point where it develops a simple reordering strategy and the subsequent modules (i.e., the encoding and decoding module in Figure 1) quickly adapt to naive reorderings. For example, in the EN-JA translation task, the reordering module tends to completely invert the input English translation after training. While this simple strategy proves to be a useful heuristic [4], we would like more accurate reordering to emerge during training. This issue is similar to posterior collapse [5], a common issue in training variational autoencoders.

Inspired by He et al. [6], we speculate that the issue occurred due to that optimization of the structured reordering module usually lags far behind the optimization of subsequent modules during the initial stages of training. We use a simple training strategy to alleviate the issue. Specifically, during the initial $M$ training steps, with a certain probability $p$, we only update the parameters of

| Name | Range |
|---|---|
| embedding size | [128, 256, 512] |
| number of encoder LSTM layer | [1,2] |
| encoder LSTM hidden size | [128, 256, 512] |
| decoder LSTM layer | [1,2] |
| decoder LSTM hidden size | [128, 256, 512] |
| decoder dropout | [0.1, 0.3, 0.5, 0.7, 0.9] |
| temperature of Gumbel-softmax | [0.1, 1, 2, 10] |
| label smoothing | [0.0, 0.1] |

Table 1: Main hyperparameters of *ReMoto*.

the structured reordering module and ignore the gradients of the parameters from the subsequence modules. $M$ and $p$ are treated as hyperparameters. With this strategy, the structured reordering module is updated more often than the subsequent modules, and has a better chance to catch up with the optimization of subsequent modules. We find that this simple training strategy usually leads to better segment alignments and better performance.