# OpenReview forum: "Structured Reordering for Modeling Latent Alignments in Sequence Transduction"
_NeurIPS.cc/2021/Conference — NeurIPS 2021 Poster_

### Official Review · Reviewer_VWV9 · 2021-07-10

**Rating:** 7
**Confidence:** 4

**Summary:**

The authors propose a model for sequence transduction tasks that allows for source tokens to be reordered and then monotonically aligned with the target tokens. The reordering and the alignment are latent, and since all monotonic alignments can be marginalized over using the approach introduced in the SSNT paper (Yu et al., 2016), the authors focus on approximating the marginalization over the latent source permutation. The authors consider in particular separable permutations, whose derivations are induced by a PCFG. The authors propose two approximation approaches, one which uses a dynamic program to obtain an expected permutation matrix (which is used instead of the expected likelihood under the permutation-matrix distribution), and one which samples a hard permutation matrix during learning and uses a ST-Gumbel-Softmax gradient estimator. The authors experiment on a number of synthetic and small-scale real-world tasks and generally find that their approach outperforms baselines, and is especially performant in settings that require generalization to longer inputs than those seen in training.

**Limitations And Societal Impact:**

Yes.

**Main Review:**

The paper is interesting, generally clearly written, and it seems technically sound. I think the core contribution of the paper is, as the authors suggest on line 66, to propose a hierarchical approach to modeling permutations of input, as opposed to other, flatter alternatives.

Overall I think this is a nice paper, but I have some suggestions/concerns:
- I'm not sure the authors' emphasis on the model capturing segments or on segment alignment is either justified or needed. As far as I can tell, the SSNT model computes alignments between tokens. It's true that segmentations are induced either by multiple tokens aligning to a single token or by the alignment skipping contiguous source tokens, but it seems like the model doesn't ever really represent segments themselves as first-class objects. Rather, it seems like the major inductive bias afforded by SSNT-type models is that the alignment needs to be monotonic, and so an alternative hypothesis for the success of both SSNT and ReMoto is that monotonic (token-level) alignment is a useful inductive bias (even if perhaps we need the alignment to be monotonic in a reordered source sequence). So, while I don't think this hurts the paper much, I think we would need much more evidence before concluding it's the segment alignment that is helping here.
- Experimentally, I think the paper is generally good, although it seems important to have an SSNT baseline in Table 3 for the semantic parsing experiments (so we can separate the contributions of monotonic alignment and reordering).
- Given that Hard-ReMoto seems to generally work best, it might be worth discussing to what extent we need the Markov assumptions made by the permutation-derivation model. For instance, would it better to just use an autoregressive parameterization of the derivation distribution?
- There are other approaches to obtaining differentiable permutations that should perhaps be mentioned, like " Stochastic optimization of sorting networks via continuous relaxation" (ICLR, 2019) and "Differentiable Ranks and Sorting
using Optimal Transport" (NeurIPS 2019)

Minor suggestions:
- I think the authors should explicitly describe the SSNT model and the parameterization of p(y | X, M) in the main text; otherwise it is hard to follow what exactly is going on.
- Equation (9) should probably have just p_{phi} rather than an expectation under it.

**Time Spent Reviewing:**

5

---

> ### Author Response · Authors · 2021-08-10
> **Response to Reviewer VWV9**
>
> We sincerely thank the reviewer for the constructive feedback.
>
> ## 1. Whether modeling segment-level alignment is justified
>
> > “I'm not sure the authors' emphasis on the model capturing segments or on segment alignment is either justified or needed”
>
> Thanks for bringing up this important point!
> We agree that, differently from our proposed structured reordering module, the top SSNT module does not represent segments as first-class objects for learning. However, **we need to highlight that segment boundaries, which are the primitive elements that are modeled by SSNT-type modules [1,2] for monotonic alignments, are conceptually and practically different from standard token-level alignments.** For example, if we were to generate $b_1$ $b_2$ from $a_1$ $a_2$ $a_3$, SSNT is expected to emit $b_1$ and $b_2$ from the segment boundary $a_3$ after digesting $a_1$ $a_2$. Standard token-alignment modules presumably need to assign alignments heuristically, say $a_1$ to $b_1$, $a_2$ to $b_2$, $a_3$ to $b_2$. Such distinction is profound in cases where we need to inject some lexicon constraints, i.e., segment-to-segment dictionaries. We will clarify this.
> > “ I think we would need much more evidence before concluding it's the segment alignment that is helping here”
>
> We will clarify that the way segment alignments are modeled in ReMoto are not ideal in a sense as segments are indirectly handled by segment boundaries, rather than directly treated as primitive objectives.
>
> ## 2. Markov assumption
>
> > “would it better to just use an autoregressive parameterization of the derivation distribution?”
>
> Thank you for the interesting question.
> We think that autoregressive parameterization is indeed a potential extension. But without any factorization, the only option for learning is to sample a sequence of local decisions (either relaxed or not), which is probably not trivial to work in practice [e.g., 3]. Moreover, the tractability resulting from the Markov assumption makes it possible to inject other structural biases such as sparsity  [4,5] which require exact MAP or top-K inference.  We will leave a rigorous study of this question to future work.
>
> ## 3. SSNT baseline for semantic parsing
>
> > “it seems important to have an SSNT baseline in Table 3 for the semantic parsing experiments”
>
> Thanks for the suggestion.
> The results for the Chinese semantic parsing are listed as follows. SSNT underperforms Hard-ReMoto in all three splits, implying that the proposed reordering module is indeed a crucial component for this task. We will add the full results in the final version.
>
>
> | Model | IID | TEMP | LEN |
> |----|-----|----|----|
> |Seq2Seq| 72.5 | 25.4| 19.8|
> |Syntactic Attention|70.2| 27.9| 18.7|
> |SSNT | 71.6 | 23.8 | 17.8 |
> |Soft-ReMoto | 73.4  | 30.3 |17.3 |
> | Hard-ReMoto | 74.3 | 45.7 | 22.3 |
>
>
> ## 4. Others
>
>
> > “...other approaches to obtaining differentiable permutations”
>
> Thank you for the references. We will discuss the papers.
>
> > “ the authors should explicitly describe the SSNT model and the parameterization of p(y | X, M) in the main text”
>
> Thank you for the suggestion! We will add more details on SSNT in the final version (where one extra page is permitted).
>
> > “Equation (9) should probably have just p_{phi}”
>
> Thank you for pointing out the writing mistake. We actually use a greedy strategy, so the correct form of Eq9 should be
> 	$$argmax_y \ \ p_{\theta, \phi'}(y|M^{D’}_{pe} X)$$
>
> where $D’ := argmax_{D} \ \ p_\phi(D|x)$.
>
> ### References
> 1. Online segment to segment neural transduction
> 2. Sequence modeling via segmentations
> 3. Cooperative Learning of Disjoint Syntax and Semantics
> 4. SparseMAP: Differentiable Sparse Structured Inference
> 5. Efficient Marginalization of Discrete and Structured Latent Variables via Sparsity

---

### Official Review · Reviewer_pMgt · 2021-07-12

**Rating:** 6
**Confidence:** 3

**Summary:**

The order of the input and output (including their associated representations) can matter for the performance and interpretability of a sequence to sequence model. Based on monotonic alignment seq2seq model (e.g. SSNT), the paper proposes to learn a separable permutation over the input representations so that the monotonic alignment can lead to better performance and interpretability. The authors test their methods on toy datasets and small scale semantic parsing and medium sized machine translation tasks.

**Limitations And Societal Impact:**

I think the authors could mention in the future work testing the methods on large scale datasets (for both semantic parsing and machine translation). Also, the proposed methods seem to integrate well with current pretrained methods and are worth trying in the future to boost task performance.

**Main Review:**

Monotonic alignment seq2seq model is an interesting family of seq2seq models that can not only boost performance (particularly when the monotonic assumption holds in large) but also provide some interpretability for the prediction task. To relax the monotonic assumption, this paper proposes to learn a permutation matrix $M_p$ as a first step before the monotonic alignment seq2seq model; the paper proposes a way to combine these two steps in an end to end fashion.
The permutation matrix is parameterized only for separable permutations, which have their linguistic groundings. In 3.2, the authors show that the matrix can be estimated efficiently as an expectation of the underlying BTG separable permutation marginals. An alternative is to add a gumbel softmax trick to simulate the hard permutation choice. The separable permutation marginal calculation is new and the authors have shown the proofs and important technical details for this algorithm (e.g. the WCFG normalization). The algorithm is generally applicable to any parameterized BTG, however neither in the paper nor in the appendix, I can see how the BTG is parametrized for various experiments, making myself unable to reproduce the experiments in thoughts.

The authors have shown the benefits of the proposed approach via toy examples and some real tasks. One question I have for the Geoquery experiments is that the baseline and the proposed methods have too low accuracy (maybe due to missing argument mapping), as a reference, https://arxiv.org/pdf/1601.01280.pdf reports a seq2seq with attention baseline, achieving 84.6 accuracy on Geoquery, and this is not SOTA at all.

The other minor writing/formatting/missing references:
- Eq(2), this formulation is not used throughout the paper and potentially misleads readers to factorization methods. From my understanding, the paper proposes two steps approaches and the matrix doesn't necessarily come from the factorization in Eq(2).
- Line 155: It seems that the order of the two non terminals should be reversed.
- Line 226: Gumbel-Softmax should be outside of parenthesis.
- In general, the order of the input matters for performance which can be another motivation of the paper, thus related to https://arxiv.org/pdf/1511.06391.pdf. The paper also connects to self attention widely used in NLP, it seems that specific parametrization of self attention can lead to permutations.

＃After discussions with the authors: the authors have clarified all my questions concerning the formalism, presentation and experiments. I feel that the paper can benefit from discussing in details the various settings and the rationales behind, not only the semantic parsing experiments that we have discussed here but also other scenarios as other reviewers pointed out. The authors have shown generally deep understanding of the encountered problems and I have adjusted my score from 5 to 6, confident that the final paper will be better quality to best convey the proposed ideas.

**Time Spent Reviewing:**

3.5

---

> ### Author Response · Authors · 2021-08-10
> **Response to Reviewer pMgt**
>
> We sincerely thank the reviewer for the detailed and helpful review.
>
> ## 1. BTG parameterization
>
> > “however neither in the paper nor in the appendix, I can(not) see how the BTG is parametrized for various experiments, making myself unable to reproduce the experiments in thoughts.”
>
> Thanks for pointing this out. Rules in our BTG are scored independently. To expand on our brief description in the appendix (Architecture and Hyperparameter paragraph), the score function for each rule has form $G(R_{i,j,k}) = MLP([s_{i,j}, s_{j,k}])$, where $s_{i,j}$ is the span embedding based on [1], $MLP$ is a multi-layer perceptron that outputs a 2-d vector, which corresponds to the score of R=Straight and R=Inverted, respectively.  We will add more details on the parameterization.
>
> ## 2. Whether our baseline model is weak
>
> > “for the Geoquery experiments is that the baseline and the proposed methods have too low accuracy”
>
> Thanks for raising the concern.
> **The major difference between our work and [2] does not lie in the models per se, but in the form of the programs.** Specifically, they used lambda calculus whereas we use a parentheses-free functional form which has a different correspondence to natural language. It’s acknowledged in the semantic parsing community that the form of programs has a significant impact on accuracy [3]. **To show this in our case, we experimented with our seq2seq baseline with the lambda calculus used by [2], and we obtain around 85% accuracy, which is comparable with their numbers [2].**  Also, as mentioned by the reviewer, they manually detect arguments during preprocessing, making the modeling task easier.
>
> We would like to emphasize that most state-of-the-art semantic parsers in the NLP literature are specialized for semantic parsing. For example, they often feature a tree-based decoder to take advantage of the readily-available grammar of programs, but such a decoder is not applicable for machine translation. In contrast, the focus of our work is a generic seq2seq model that is applicable for both semantic parsing and machine translation.
>
> ## 3. Others
>
> > “Eq(2), this formulation is not used throughout the paper and potentially misleads readers to factorization methods.”
>
> We are not sure if we completely understand the concern. We would like to clarify that Eq2 is a general formulation and serves as a backdrop against which to interpret the more concrete formulation in  Eq3.  We will clarify that we are not using matrix factorization methods.
>
>
> > “Line 155: It seems that the order of the two non terminals should be reversed.”
>
> **Intuitively, the rules that compose the BTG derivation tree (e.g., Figure 3) only signify  which segments to inverse**. It is during the process where a tree is mapped to a permutation matrix (as introduced in 3.2) that the label (i.e., straight or inverted) is reflected by the choice of matrix operation. We will clarify this.
>
> > “the order of the input matters for performance which can be another motivation of the paper”
>
> Thanks for pointing out the potential connections. We will add them in the revised version.
>
> ### References
> 1. Graph-based dependency parsing with bidirectional lstm
> 2. Language to Logical Form with Neural Attention
> 3. Benchmarking Meaning Representations in Neural Semantic Parsing

---

> > ### Comment · Reviewer_pMgt · 2021-08-11
> > **Thanks for the detailed response, some further questions**
> >
> > Thank you for the detailed response.
> >
> > If the paper is accepted, concerning my review I would encourage the authors to: 1) briefly talk about the BTG parameterization 2) explain more on line 154-155 (as both the first reviewer and myself are confused by this grammar first reading it) 3) reformulate/clarify that eq2 is just a backdrop (my original point is simply that I don't think this equation is needed as eq3 seems pretty neat itself)
> >
> > My only remaining confusion is that why authors don't simply report the experimental results using lambda calculus as it is a more widely used formalism across various popular systems [1,2]? If the lambda formulation tends to violate the monotonic assumption (even after permutation) so the authors choose the variable-free formalism, please just state and discuss this clearly in the paper.
> >
> > [1] https://github.com/donglixp/coarse2fine
> > [2] https://github.com/pcyin/tranX

---

> > > ### Author Response · Authors · 2021-08-13
> > > **Regarding the choice of program formalisms for semantic parsing**
> > >
> > > Thank you for the suggestions and the further question.
> > >
> > > There are two reasons for using a variable-free formalism here. First, in this work, we primarily focus on compositional generalization (CG) setting, and previous work on CG in semantic parsing used variable-free formalisms [1,2]. They did it probably because it is easier to construct generalization splits with such representations (e.g., defining templates or MCD splits). Second, as you also pointed out, variable-free representations have more straightforward correspondences with natural language than Prolog and lambda calculus.  For this reason, in semantic parsing literature, variable-based formalisms have been commonly used in alignment-driven approaches [3,4]. Intuitively, variables in programs make alignments hard to define. For example, in the language-program pair “what states border Texas? -> lambda x. state(x) ^ border (x, texas)”, it would be tricky to define the correspondence of the program segment “lambda x” in a way for an alignment-based model to learn. We will clarify this.
> > >
> > > ### References
> > > 1. Span-based semantic parsing for compositional generalization
> > > 2. Compositional generalization and natural language variation: Can a semantic parsing approach handle both?
> > > 3. Learning for Semantic Parsing with Statistical Machine Translation
> > > 4. Semantic Parsing as Machine Translation

---

### Official Review · Reviewer_iq7F · 2021-07-20

**Rating:** 8
**Confidence:** 3

**Summary:**

The paper tackles the problem that conventional sequence-to-sequence models (seq2seq models) fail to generalize systematically; i.e., they are suboptimal for handling the compositionality of language. The paper makes the following contributions: (1) a new seq2seq model for NLP tasks that accounts for latent non-monotonic segment-level alignments and (2) an algorithm for exact marginal inference with separable permutations, which makes end-to-end training possible. The proposed approaches are evaluated both on synthetic data and two real-world NLP tasks: semantic parsing and machine translation. The experiments show that the proposed model mostly outperforms conventional seq2seq models in an IID setting, and even more strongly outperforms existing models in settings where test examples are longer than training examples.

**Ethical Concerns:**

None.

**Limitations And Societal Impact:**

The authors do not explicitly talk about the limitations of their work, but the experiments make some potential limitations obvious (e.g., seq2seq could perform better in some IID settings). I don't see a reason to expect any negative societal impact of the work.

**Main Review:**

The authors do a great job introducing the motivation for their work (i.e., the problems with currently existing models). Furthermore, the experiments are designed well to show the benefits of the proposed approaches, and I like the fact that the experiments are done on both synthetic data and real-world datasets. The experimental results demonstrate the benefits of the proposed model. The paper is written clearly.

**Time Spent Reviewing:**

2

---

> ### Author Response · Authors · 2021-08-10
> **Response to Reviewer iq7F**
>
> Thanks for your review. We are greatly encouraged by your positive comments!

---

### Official Review · Reviewer_fSuH · 2021-07-21

**Rating:** 7
**Confidence:** 3

**Summary:**

This paper targets the problem of systematic generalization in sequence to sequence models, particularly in the scenario of modeling segment alignments as discrete structured latent variables. To explore the searching space of alignments, the authors proposed to use a reorder-first align-later framework, and use hierarchical permutation trees to produce separable permutations, which has the exact marginal inference with dynamic programming enabling end-to-end training. Experiments show that the proposed approach outperforms standard models on both synthetic and real NLP tasks with better systematic generalization.

**Limitations And Societal Impact:**

The author did not have a separate section about the limitation or societal impact.

**Main Review:**

Overall I think the idea of reordering and then model alignment is quite interesting and important for the area of seq2seq learning, as it solves a big issue of conventional approaches (e.g. SSNT) that only works for monotonic alignment while in real scenarios many applications such as machine translation do not hold. Also, the motivation of connecting learning systematic generalization and segment alignments seems reasonable.

I only have some detailed question about this submission:

-- In section 2.2, you discussed the connection between discrete alignment and continuous attention. Will the proposed target also be achieved by using the attention weights for the segment alignments? Or is there any reason we have to work on discrete alignment (which might potentially be more costly)?

-- In Line 154 & 155, do these two equations have a typo? Why do “straight” and “inverted” have the same results? Moreover, it would be much better to give more context explaining these terms such as “straight”, “inverted” as well as BTG to make it easier to understand the method.

-- What is the algorithm you used for inference? How to decide the best permutation of reordering?

-- All the experiments seem to be quite small scale with a small architecture. Is it because the proposed method cannot work on a larger scale? Will this idea be applied to large scale problems such as WMT using Transformer based models?

-- What is the computational cost for both soft and hard ReMoto, and how does this compare to previous works in both training and inference time?


**Time Spent Reviewing:**

3-4 hours

---

> ### Author Response · Authors · 2021-08-10
> **Response to Reviewer  fSuH**
>
> We sincerely thank the reviewer for the constructive feedback.
>
> ## 1. Connection between continuous attention and discrete alignments
>
> > “Will the proposed target also be achieved by using the attention weights for the segment alignments? Or is there any reason we have to work on discrete alignment (which might potentially be more costly)?”
>
> This is an interesting question!
> **Our soft reordering version can actually be regarded as an instance of an attention network**, despite being defined in a structured way. Specifically, although motivated by discrete alignments, our soft reordering results in a continuous matrix  M’_pe  which can be viewed as a matrix of structured attention [1], and a replacement of standard "unstructured" attention. In this case, discrete alignment only provides a probabilistic interpretation of such structured continuous attention, similarly to [1].
>
> **However, continuous relaxation often leads to suboptimal performance.** Specifically, the relaxation from discrete alignment to continuous attention results in a significant change in the training objective (i.e., from Eq 4 to line 188), and such change leads to a drop in performance:  as we show in Experiments,  systems based on hard reordering outperform those based on soft reordering in most of our experiments. In general, directly maximizing the marginal likelihood based on discrete alignments seems to be more desirable than continuous relaxations, as observed also in other work, such as [2,3].
>
> ## 2. BTG grammar rules
>
> > “Why do “straight” and “inverted” have the same results?”
>
> Intuitively, **the rules that compose the BTG derivation tree (e.g., Figure 3) only signify  which segments to inverse**. It is during the process where a tree is mapped to a permutation matrix (as introduced in 3.2) that the label (i.e., straight or inverted) is reflected by the choice of matrix operation. We will clarify this.
>
> > “it would be much better to give more context explaining these terms such as “straight”, “inverted” as well as BTG”
>
> Thanks for the suggestion, we will add more explanation on BTG in the revised version.
>
> ## 3. Inference
>
> > “What is the algorithm you used for inference?”
>
> During inference, we use a greedy strategy: 1)  find the most probable reordering, 2) obtain the best output given the reordering.  We briefly mentioned this in Eq 9, and we will make this more clear in the next version.
>
> > “ How to decide the best permutation of reordering?”
>
> The inference algorithm to find the most probable reordering is a straightforward adaptation of the sampling procedure in Algorithm 1, by changing stochastic s_argmax in line 8 to a deterministic argmax.  We will add the description.
>
> ## 4. Efficiency and Scalability
>
> > “What is the computational cost for both soft and hard ReMoto”
>
> In semantic parsing, our models (both soft- and hard- ReMoto) increase the training and inference time by a factor of 2; in machine translation, the factor is 4-6, compared with a standard seq2seq baseline. We will discuss this in our revised version.
>
> > “All the experiments seem to be quite small scale with a small architecture.”
>
> Our choice of small scale datasets is indeed due to computational costs. However, we should point out that more than 70% computations (measured by execution time of modules) are caused by the SSNT module. **Hence, to perform large-scale experiments with ReMoto, the bottleneck is not our proposed reordering module. Rather, a more efficient alternative for accommodating monotonic alignments can be used.** As a preliminary attempt, we have replaced SSNT with CTC [4], which also enforces monotonicity and has a flavor of non-autoregressiveness. The resulting ReMoto variant is very efficient (even slightly faster than a conventional seq2seq baseline) and performs even better than Hard ReMoto in semantic parsing, but does not perform as well in machine translation. We will add discussion to this end and call for research on a more scalable and effective model for monotonic alignments.
>
> ### References
> 1. Structured Attention Networks
> 2. Show, Attend and Tell: Neural Image Caption Generation with Visual Attention
> 3. Latent Alignment and Variational Attention
> 4. Connectionist Temporal Classification: Labelling Unsegmented Sequence Data with Recurrent Neural Networks

---

### Decision · Program_Chairs · 2021-09-27

**Decision:**

Accept (Poster)

**Comment:**

This paper presents uses the class of separable permutations, which in contrast to the unrestricted class of permutations, can be reasoned about in polynomial time using dynamic programming algorithms to introduce explicit reordering and alignment variables in seq2seq models. The reviewers remarked on the technical clarity, interestingly novel approach, and thorough experiments. This technique improves interpretability and, most importantly, provides a demonstrated bias for compositional generalization, both of which are important concerns in sequence transduction modelling. This represents a serious and successful attempt to address these issues by changing the underlying assumptions of the model, rather than relying on data augmentation.